# Automating the amino acid identification in elliptical dichroism spectrometer with Machine Learning

**Ridhanya Sree Balamurugan**[1], **Yusuf Asad**[1], **Tommy Gao**[2], **Dharmakeerthi Nawarathna**[3], **Umamaheswara Rao Tida**[1]*, **Dali Sun**[2]

**1** Electrical and Computer Engineering, North Dakota State University, Fargo, North Dakota, United States of America, **2** Electrical and Computer Engineering, University of Denver, Denver, Colorado, United States of America, **3** Electrical and Computer Engineering, Old Dominion University, Norfolk, Virginia, United States of America

* umamaheswara.tida@ndsu.edu

**Data Availability Statement:** Data for this article, including the codes built to analyze, are available at {https://github.com/Ridhanya/Amino_Acid_Classification_ED}.

## Abstract

Amino acid identification is crucial across various scientific disciplines, including biochemistry, pharmaceutical research, and medical diagnostics. However, traditional methods such as mass spectrometry require extensive sample preparation and are time-consuming, complex and costly. Therefore, this study presents a pioneering Machine Learning (ML) approach for automatic amino acid identification by utilizing the unique absorption profiles from an Elliptical Dichroism (ED) spectrometer. Advanced data preprocessing techniques and ML algorithms to learn patterns from the absorption profiles that distinguish different amino acids were investigated to prove the feasibility of this approach. The results show that ML can potentially revolutionize the amino acid analysis and detection paradigm.

## 1 Introduction

Detecting amino acids is of utmost importance as they are the foundational building blocks in synthesizing proteins and play crucial roles in metabolic pathways, nutritional deficiencies, drug development and disease detection [1, 2]. Many traditional amino acid detection methods exist, such as mass-based spectrometry and optical-based spectrophotometry [3]. Mass-based spectrometry function on the weight of molecules in a sample and report more accurate and reproducible results however often they are coupled with separation techniques such as liquid chromatography, thin layer chromatography, or gas chromatography [4] leading to extensive sample preparation, time-consumption, high complexity, and cost [5]. Optical based spectrophotometry based on absorbance of light through a sample, due to detection by use of photodetector does not offer as accurate and reproducible results but instead can offer distinction between more general structural motifs of biomolecules such as $\alpha$-helix and $\beta$-sheet secondary structures of proteins [6, 7]. As seen in circular dichroism (CD) spectrophotometry, it can distinguish handedness (chirality) states of amino acids with aid of some molecular fluorescent probes attached to biomolecules of interest [8]. Thus, for use where high resolution complete biomolecule complexity is not required, optical based spectrophotometry can be

**Funding:** This work was financially supported by grants from the National Cancer Institute (R21CA270748, R03CA252783) and the National Institute of General Medical Sciences (U54GM128729) of National Institutes of Health to Dali Sun, National Science Foundation CAREER Award (2236885) to Dali Sun and National Science Foundation PFI-TT Award (2300064) to Dharmakeerthi Nawarathna.

**Competing interests:** The authors have declared that no competing interests exist.

used for amino acid detection while still offering benefits of less sample preparation time when compared to mass-based techniques.

CD spectroscopy, a traditionally standard way of biomolecule detection and analysis used currently, offers high-resolution and accurate analysis of molecule chirality in a sample using circularly polarized light [9]. CD spectroscopy also has many different techniques to analyze biomolecular structures such as electronic circular dichroism [10]. Fundamentally, when left and right circularly polarized light is passed through a sample, the light is absorbed differently depending on incident light to the biomolecule's bonds and structure [11] thus offering chirality distinction. However, due to the intricate parts required to generate circularly polarized light, namely high-powered laser, photoelastic modulator (PEM), monochromator, and often cooled photodetector, size and cost are high for CD spectrophotometers and spectrometers in general [12]. Functioning on the same principle as CD spectrophotometry where circularly polarized light is incident upon a sample, a new approach was derived by using elliptically polarized light which is markedly easier to generate [13].

The design of this elliptical dichroism spectrophotometer (EDS) is detailed in our previous publication [13]. In short, instead of the costly PEM used in CD spectrometry, this EDS generates elliptically polarized light by using a collimator with an iris and a quarter waveplate mounted inside a motorized rotator [13]. Furthermore, instead of filtering out a specific wavelength of light for the absorbance reading as seen in CD spectrophotometry, a broadband UV deuterium lamp with a working wavelength range between 185–400nm (Hamamatsu) is used in the EDS. This design of the EDS tackles some of the issues that traditional biomolecular analysis instruments face while still offering sufficient resolution and analysis of a given sample. Additionally, the EDS design is much simpler while offering low-cost portability features not found in CD spectrophotometers [13].

Optical-based spectrometers, like the EDS, function based on the absorbance of light through a sample [14]. Since different biomolecules have different amino acid composition, folding, and structural features [15], they exhibit different absorbance values. Due to the use of a motorized rotator in the EDS, absorbance readings are taken at angle increments of 5 degrees through a full 360 degrees of rotation [13]. These absorbance values at specific angles (absorption profile) are unique to the sample exhibiting key biomolecular features. Utilizing these absorption profiles from the EDS, we introduce a pioneering Machine Learning (ML) approach for automatic amino acid identification in this paper.

Given the phenomenal success of ML in various tasks like object detection [16], image classification [17], and segmentation [18], our research aims to study the capabilities offered by contemporary ML techniques in amino acid detection. By developing an ML model grounded in robust spectral data, we endeavour to achieve a level of precision and efficiency hitherto unattainable with conventional methods. Although ML has been investigated for biomolecular [19] and mass spectrometry data [20], to the best of our knowledge, there are no previous works that focussed on using ML for amino acid classification using EDS data. Therefore, our research stands first in investigating this approach.

The main aim is to develop an ML algorithm that can automatically detect the type of amino acid from the absorption profile generated from the EDS for a particular unknown amino acid sample. To develop such an algorithm, in this paper, we first discuss the detailed protocol we followed to acquire the necessary data from the EDS to conduct the analysis. Then, we elaborately discuss the preprocessing methods and different ML algorithms investigated. Finally, towards the end, we present the results of our analysis, where we showcase the performance of different investigated ML algorithms in amino acid classification.

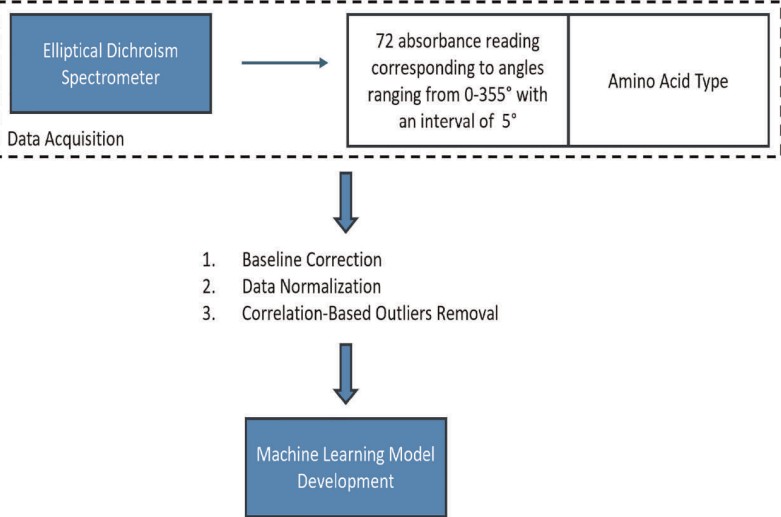

**Fig 1. Overall demonstration of methodology.**

## 2 Methodology

Fig 1 demonstrates the steps we followed to investigate ML for automatic amino acid identification. The first step involved acquiring data to study different ML models, where we used the EDS to collect the absorption profiles of samples of known amino acid type. These acquired absorption profiles must be collected to train ML algorithms, enabling them to learn patterns that distinguish different amino acid types. Next, we preprocessed the acquired data to enhance the performance of investigated ML models. In this step, we applied three preprocessing techniques: Baseline Correction, Data Normalization and Correlation-Based Outliers Removal. This was followed by investigating several popular ML algorithms to automate amino acid type identification. These steps are detailed in the sections below.

### 2.1 Data acquisition

In order to train an ML algorithm to detect the right amino acid type in the EDS automatically, we require the absorption measurements from the EDS for the known amino acid samples. To collect such information, amino acids in lyophilized powder form were dissolved in phosphate-buffered saline (PBS, pH 7.0) to a final concentration of 1 milligram per millilitre. 200 microliters of the sample solution were loaded into a clean cuvette with a stopper lid and inserted into the cuvette holder in the EDS. Before beginning measurements, all electrical components of the EDS were turned on for at least 30 minutes to ensure stable readout. The predominant source of noise is the cost-effective photodetector, selected for its simplicity and clinical applicability, with optical components securely mounted to mitigate mechanical perturbations. In the EDS operating mode, absorbance measurements are taken in 5 degree increments through a full 360 degree range. This produces absorbance profiles known as absorbance sweep graphs, which are then used for further preprocessing techniques and ML. Additionally, blank measurements conducted under identical experimental conditions ensure robust normalization, thereby isolating intrinsic amino acid spectral features irrespective of solvent or pH variations. Also, post-normalization noise levels remain competitive, with the ED spectrometer exhibiting a consistent signal-to-noise ratio (SNR $\approx$ 14), as demonstrated in S1 Fig.

**Table 1. Data distribution across 17 amino acid types.**

| Amino Acid Type | Data Count |
| --- | --- |
| L-Alanine | 132 |
| L-Arginine | 142 |
| L-Asparagine | 17 |
| L-Cystine | 12 |
| L-Glutamate | 26 |
| L-Glutamine | 22 |
| L-Histidine | 22 |
| L-Isoleucine | 37 |
| L-Leucine | 37 |
| L-Lysine | 42 |
| L-Methionine | 17 |
| L-Phenylalanine | 37 |
| L-Proline | 22 |
| L-Serine | 17 |
| L-Threonine | 17 |
| L-Tryptophan | 12 |
| L-Valine | 17 |
| Total | 628 |

We collected about 628 absorption profiles of 17 soluble amino acid types in this way. Table 1 shows the distribution of the data collected across the 17 amino acid categories. It is evident from this table that the data distribution is skewed, with L-Alanine and L-Arginine having the highest number of absorption measurements. Each absorption profile contains 72 absorbance measurements captured at varying angles ranging from 0˚ to 355˚ with an interval of 5˚. It is worth noting that the absorption values making these absorption profiles are derived from the logarithmic ratio of light intensities and are dimensionless quantities.

## 2.2 Data preprocessing techniques

**2.2.1 Baseline correction.**   We applied baseline correction to each collected absorption profile to eliminate background noise and systematic biases. A linear line connecting the absorbance measurements(x) at angles(y) 0˚ and 355˚, both converted to radians, was constructed for each absorption profile. The absorbance experimentally measured at varying angles(y) ranging from 0˚ to 355˚ with an interval of 5˚ in these absorption profiles were then subtracted from the corresponding x values from their respective linear baselines. Fig 2(a) and 2(b) demonstrate the trend of an absorption profile of L-Alanine before and after baseline correction. The linear baseline constructed for each absorption profile can be mathematically represented as follows:

$$y = ax + b \tag{1}$$

where b = absorbance measurement at 0˚(converted to radians) and a = absorbance measurement at 355˚(converted to radians)-b/355˚(converted to radians).

**2.2.2 Data normalization.**   We normalised data to amplify the variations and adjust the absorbance readings to a common scale. In this step, the mean($\mu$) and standard deviation($\sigma$) were computed for the 72 absorbance readings in each absorption profile. Each absorbance

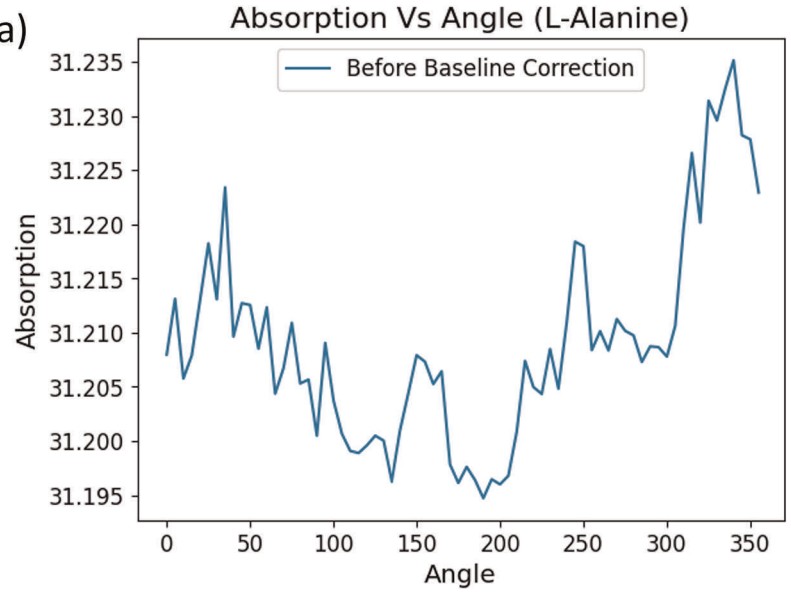

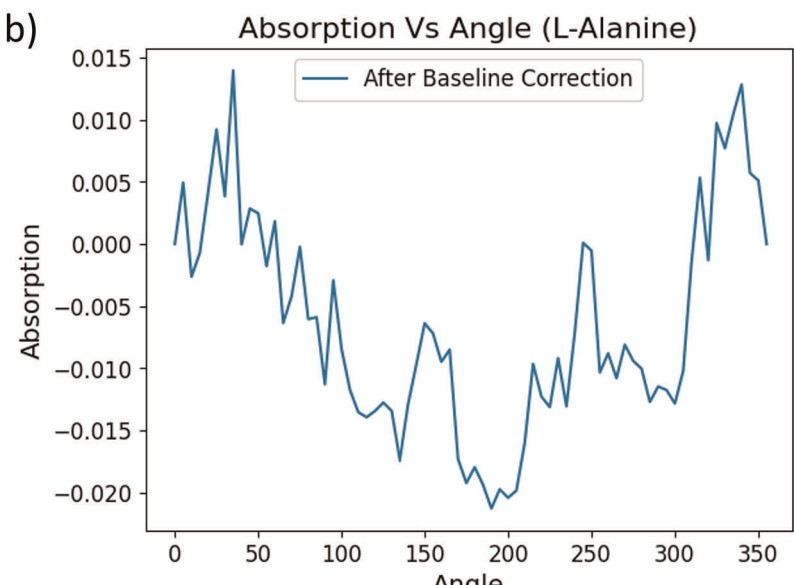

**Fig 2. Demonstration of absorption vs angle plot for an experiment with L-Alanine sample before and after baseline correction.** a) Plot of Absorption Vs Angle for an experiment with L-Alanine before baseline correction. b) Plot of Absorption Vs Angle for an experiment with L-Alanine after baseline correction.

reading in this profile was then updated using the mathematical equation below.

$$\text{Absorbance Reading} = \frac{\text{Absorbance Reading} - \mu}{\sigma} \quad (2)$$

Fig 3 demonstrates the trend of a baseline corrected absorption profile of L-Alanine before and after data normalization. Normalization enhances intra-profile variations and facilitates neural network convergence, which is crucial for stable model training and higher

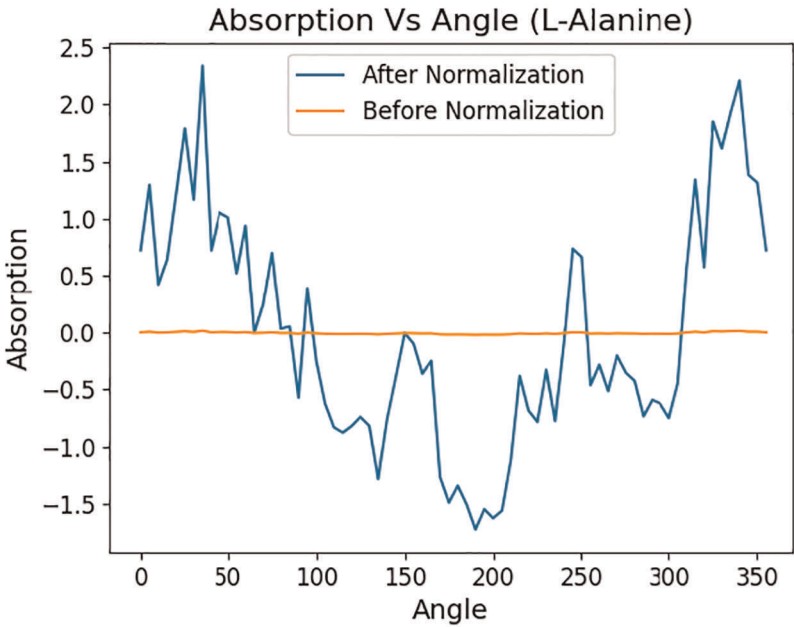

**Fig 3. Demonstration of absorption vs angle plot for an experiment with L-Alanine sample before and after data normalization.**

performance. Additionally, we included the plots of baseline corrected and normalized absorption profiles for all the 17 amino acid types, indicating standard mean error in S2 Fig.

**2.2.3 Correlation based outliers filtering.** We performed a correlation-based outliers filtering process to eliminate outliers in the acquired data, which can potentially affect the performance of ML models. In this step, the absorption profiles, experimentally collected for each amino acid type, were utilized to compute a mean sequence. Like an absorption profile, the mean sequence consists of absorbance readings at varying angles ranging from 0° to 355° with an interval of 5°. For an amino acid type, each of these readings is the mean of absorbance measurements of the respective collection of absorption profiles at each varying angle ranging from 0° to 355° with an interval of 5°. That is, for an amino acid group, the mean sequence's absorption reading at angle 0° is the mean of absorption readings of the respective collection of absorption profiles at angle 0°. In this way, for 17 amino acid types, 17 mean sequences were computed and used to find their correlation (Spearman) with the respective groups of adsorption profiles. Three absorption profiles that were least correlated with their corresponding mean sequence were removed for further analysis. This approach of Spearman correlation-based outlier removal provides a balance between computational efficiency and data quality; subsequent research will investigate the utilization of Principal Component Analysis (PCA) and Uniform Manifold Approximation and Projection (UMAP) for enhanced data visualization.

## 2.3 ML algorithm development

After the necessary data preprocessing procedures, we used the preprocessed absorption profiles of known amino acid samples to investigate traditional ML algorithms like Random Forest [21], Decision Tree [22] and XgBoost [23]. We also delved into designing neural networks optimized for performance. This paper's Results and Discussion section elaborates on this neural network algorithm's design.

As each of the absorption profiles is associated with an amino acid category, which acts as a class label that is essential for training a classification ML model to predict the correct class for new, unseen data, proper encoding techniques were applied to ensure that these class labels were converted into a format that ML algorithms understand. For this purpose, one-hot encoding of class labels was performed to train neural networks and label encoding of class labels was performed to train traditional ML algorithms.

One-hot encoding, implemented using Pandas 1.5.1, transforms each class label into a binary vector representation. For instance, to classify between 17 amino acid types in our investigation, we created a binary vector of length 17 for each absorption profile. The element corresponding to the class label of the absorption profile was set to 1, while all other elements were set to 0. This element, represented with a 1, indicates the presence of the associated amino acid category. On the other hand, label encoding, implemented using Scikit-Learn 1.1.3, transforms each class label into a numerical format. Each unique class label was assigned a numerical value ranging from 1 up to 17, denoting 17 amino acid types. Therefore, for each absorption profile belonging to a specific amino acid category, the corresponding numerical value of that category was assigned as its class label.

## 3 Results and discussions

In our experimental setup, we leveraged neural networks implemented using TensorFlow 2.11.1 alongside traditional ML algorithms implemented in Scikit-Learn 1.1.3 to classify between all 17 amino acid types. To train these algorithms, we performed a data split, where 70% of the data was used for training, 15% for validation, and 15% for testing. We specifically implemented a stratified data splitting technique to ensure that these three datasets—train, test and validation, contain an equal representation of all the amino acid categories. Since traditional ML algorithms typically do not incorporate a separate validation dataset during training, we integrated the validation and test data into a unified test set for each of these algorithms. Also, our study employed traditional ML algorithms with default settings from the Scikit-Learn 1.1.3 library to ensure a standardized and fair comparison across different models.

In all our experiments with neural networks, we used a batch size of 10 and trained this algorithm for 500 epochs. The Adam optimizer, a popular choice of optimizer in DL, was utilized to optimize the model parameters during training. Categorical Cross-Entropy loss was employed as the objective function. Also, the performance of this neural network model in different investigations, reported in the subsequent sections, is the results obtained at the epoch when the validation accuracy was at its peak. Moreover, in all these investigations with neural networks, we also used an early stopping strategy with a patient level of 50 to speed up the training process. That is, although we set the total number of epochs to be 500, internally, we track if the performance of the trained model at each epoch shows an increase in the performance on the unseen validation dataset. If the performance on this validation dataset decreases or stagnates in consecutive 50 epochs, the training stops automatically.

Additionally, our implementation incorporated class weights during training as our data distribution is skewed across 17 amino acid types, posing class imbalance. Class weights are an important aspect of ML algorithms, especially when the dataset used to train these algorithms poses class imbalance. Assigning class weights in such a scenario enables these ML algorithms to effectively learn features of even the poorly represented amino acid categories containing fewer data counts. We assigned class weights based on the inverse of the class or amino acid type frequency, with each class weight calculated as the total number of experimental records/ absorption profiles in the training dataset divided by the number of records in that class. However, as the XGBoost algorithm in scikit-learn does not provide an option for directly assigning

class weights, our exploration of this particular algorithm proceeded without incorporating them.

## 3.1 Classification performance in distinguishing 17 amino acid types

Table 2 summarizes our results in classifying all 17 amino acid types across different ML algorithms. Accuracy (Acc), rounded off to two decimals, obtained across the three data splits (train, test and validation datasets) is reported in this table. In ML, Acc is a metric used to evaluate the performance of a classification model, and it represents the ratio of correctly predicted instances to the total number of instances in the dataset. Additionally, since the neural network we investigated in our study contains dropout, the reported train Acc is the accuracy obtained in the training dataset with the saved best model used to report validation and test Acc. While training, the train Acc obtained was 0.08 (rounded off to two decimals).

The lower performance observed in our study can be attributed to the limited availability of experimental data, especially for certain amino acid categories like L-Tryptophan. This lack of adequate data poses a significant challenge for ML algorithms to effectively learn discriminative features and generalize well to unseen data. Despite setting class weights to address class imbalances, the dataset's small size severely constrains the models' learning capacity. It undermines their ability to extract meaningful patterns and relationships from the data.

Therefore, to study if utilizing ML for automatic amino acid identification is feasible, our next goal was to classify amino acids, L-Alanine and L-Arginine, which had the highest data count among all the 17 amino acid types. Moreover, although the overall performance was low, the investigated algorithms showcased higher performance in identifying these two amino acid categories, owing to the availability of relatively higher amounts of data acquired for these amino acid groups. Additionally, since there was inadequate data in most of the amino acid categories, using different neural network models with varying hyperparameters didn't improve the results of classifying all 17 amino acid types. Therefore, the best neural network model we obtained in classifying L-Arginine and L-Alanine was used to report the result in Table 2. The architectural details of this model are elaborated in the subsequent sections.

## 3.2 Classification performance in distinguishing L-Arginine and L-Alanine

Table 3 summarizes our results (in terms of accuracy rounded off to two decimals) in classifying L-Arginine and L-Alanine across different algorithms, and Table 4 summarizes the individual amino acid accuracy (rounded off to two decimals) obtained. As stated earlier, since the neural network we investigated in our study contains dropout, the reported train Acc is the accuracy obtained in the training dataset with the saved best model used to report validation and test Acc. While training, the train Acc obtained was 0.79 (rounded off to two decimals). Fig 4 illustrates the configuration of the DL model utilized in our study. The model was structured with specific hyperparameters tailored to optimize model performance. Furthermore, this experiment did not incorporate class weights as L-Alanine and L-Arginine do not

**Table 2. Accuracy obtained across different ML algorithms in classifying 17 amino acid types.**

| Algorithm | Train | Validation | Test |
|---|---|---|---|
| Decision Tree | 1.0 | - | 0.19 |
| Random Forest | 1.0 | - | 0.33 |
| XgBoost | 1.0 | - | 0.26 |
| Deep Learning Model | 0.19 | 0.19 | 0.14 |

**Table 3. Accuracy obtained across different ML algorithms in classifying L-Alanine and L-Arginine.**

| Algorithm | Train | Validation | Test |
|---|---|---|---|
| Decision Tree | 1.0 | - | 0.69 |
| Random Forest | 1.0 | - | 0.77 |
| XgBoost | 1.0 | - | 0.67 |
| Deep Learning Model | 0.86 | 0.83 | 0.78 |

**Table 4. Individual amino acid accuracy obtained across different ML algorithms.**

| Algorithm | L-Arginine | L-Alanine |
|---|---|---|
| Decision Tree | 0.69 | 0.7 |
| Random Forest | 0.81 | 0.73 |
| XgBoost | 0.69 | 0.65 |
| Deep Learning Model | 0.81 | 0.75 |

showcase class imbalance and contain an almost equal number of experimentally collected absorption profiles.

As depicted in the Fig 4, we utilized a combination of popular DL operations such as Conv1D, BatchNorm, ReLU, Maxpool1D, Flatten, Dropout and Dense layers to design the neural network model used to classify between L-Arginine and L-Alanine. Conv1D layers are specialized neural network layers designed to extract features from one-dimensional sequential data, such as time series or text. They utilize a learnable sliding 1D matrix to scan the input sequence to find relevant patterns. BatchNorm layer is used to improve the stability and performance of neural networks by normalizing the outputs of each layer. ReLU is an activation function commonly used to introduce non-linearity into the model. MaxPool1D is a

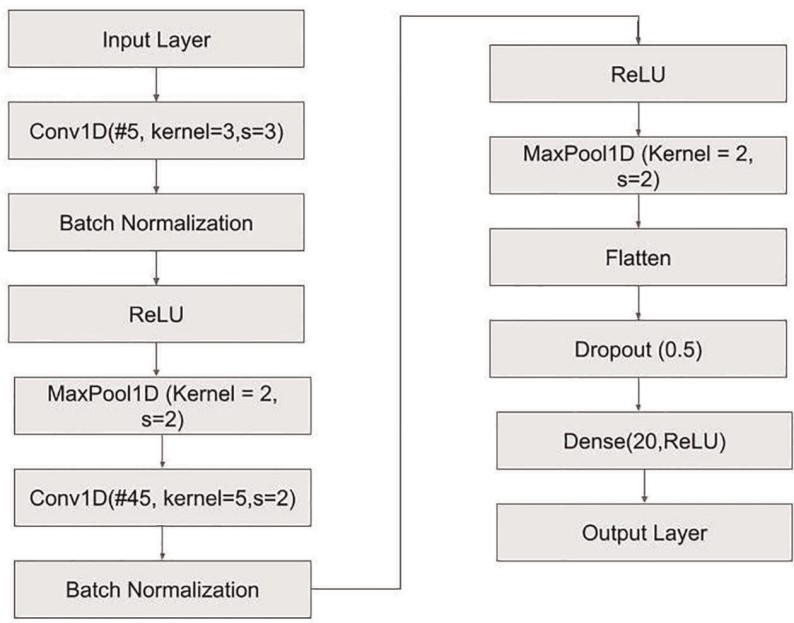

**Fig 4. Design of the neural network investigated in this study.**

downsampling operation commonly used to reduce the spatial dimensions of feature maps (or inputs) while retaining the most salient information. A Flatten layer is used to reshape a tensor to a 1D vector suitable to input to Dense Layers. Dropout Layer is used to prevent the model from overfitting, which is a scenario in which the model memorizes the train data and performs poorly on unseen test data. Dense layers are typically employed in the final stages, transforming the high-level features learned by earlier layers into predictions or classifications. Additionally, the Input Layer is where the absorption profiles are passed as input and the Output Layer is where the model outputs a prediction (or the amino acid type) for a given input absorption profile.

We manually tuned the hyperparameters such as the number of filters (#), activation functions, stride (s), kernel size and number of units in dense layer to incur the highest possible performance in distinguishing L-Alanine and L-Arginine. Additionally, we observed that decreasing or increasing the number of dense layer in combination with increasing or decreasing the number of convolution layer set—Conv1D, BatchNorm, ReLU and Maxpool1D, didn't result in increase in performance. Furthermore, we observed that increasing the number of these layers or the layer set just spiked the total number of parameters and resulted in the overfit of the model. Thus, the neural network model used to report the results in Tables 3 and 4 was chosen as the optimal model in our investigation.

From the results reported in Tables 3 and 4, it is clear that the traditional ML algorithms demonstrate overfitting. Overfitting occurs when a model learns to memorize the training data rather than capturing underlying patterns that generalize well to unseen data. It is evident from these tables that the traditional ML algorithms have higher Acc on the training dataset in comparison to the unseen test and validation datasets. However, this is not observed in the reported performances across these three datasets (train, test, and validation) of the DL model. This demonstrates that traditional ML algorithms, such as Random Forest, Decision Tree, and XGBoost, are susceptible to overfitting when trained on datasets with limited size or noisy features. Additionally, the good performance of the DL model in classifying L-Arginine and L-Alanine proves that with adequate data, DL can be leveraged to classify all 17 amino acid types in the future.

## 3.3 Classification performance in distinguishing L-Arginine, L-Alanine and L-Lysine

As we successfully classified between L-Alanine and L-Arginine, having the highest number of acquired absorption profiles in the dataset, with high performance (of about 80 percent test accuracy), we wanted to investigate the classification of L-Alanine, L-Arginine, and L-Lysine, as L-Lysine had the next highest count of absorption profiles experimentally collected. Thus, this section discusses our results in classifying these three amino acid types.

Table 5 summarizes our results (in terms of accuracy rounded off to two decimals) in classifying L-Arginine, L-Alanine and L-Lysine across different algorithms, and Table 6 summarizes the individual amino acid accuracy (rounded off to two decimals) obtained across these

Table 5. Accuracy obtained across different ML algorithms in classifying L-Alanine, L-Arginine and L-Lysine.

| Algorithm | Train | Validation | Test |
|---|---|---|---|
| Decision Tree | 1.0 | - | 0.51 |
| Random Forest | 1.0 | - | 0.68 |
| XgBoost | 1.0 | - | 0.59 |
| Deep Learning Model | 0.78 | 0.65 | 0.60 |

**Table 6. Individual amino acid accuracy obtained across different ML algorithms.**

| Algorithm | L-Arginine | L-Alanine | L-Lysine |
|---|---|---|---|
| Decision Tree | 0.47 | 0.68 | 0.08 |
| Random Forest | 0.77 | 0.78 | 0 |
| XgBoost | 0.63 | 0.73 | 0 |
| Deep Learning Model | 0.52 | 0.8 | 0.17 |

algorithms. Additionally, since the neural network we investigated in our study contains drop-out, the reported train Acc is the accuracy obtained in the training dataset with the saved best model used to report validation and test Acc. While training, the train Acc obtained was 0.64 (rounded off to two decimals). In this investigation, we incorporated class weights during training as the L-Lysine amino acid type had only one-third of the experimentally collected absorption profiles compared to the other two amino acid groups. Each amino acid group in this experiment was assigned a class weight, which was calculated as the total number of experimental records/absorption profiles in the training dataset divided by the number of records in that class.

The results show that ML algorithms perform poorly in identifying L-Lysine compared to L-Alanine and L-Arginine. The ML algorithms' inability to learn features specific to L-Lysine can be attributed to the inadequate absorption profiles collected for this amino acid type compared to L-arginine and L-alanine. The other amino acid groups, L-Alanine and L-Arginine, contain almost thrice the number of training data as L-Lysine. This further shows that ML algorithms' performance in identifying amino acids can be improved with adequate data.

## 3.4 Transfer learning approach to classify between L-Alanine, L-Arginine and L-Lysine

We further investigated if the DL model trained to classify between L-Alanine and L-Arginine can effectively learn features of L-Lysine through a transfer learning approach. Usually, in ML, when the size of the acquired dataset to train a DL model for a particular automation task is small, giving rise to poor performance in the task, we leverage the capabilities of good-performing state-of-the-art DL models trained with ample amounts of data for related ML tasks, through a process called Transfer Learning [24].

In the transfer learning approach, we freeze the weights of the architecture, except for the last few layers, of the good-performing state-of-the-art DL model, trained for an ML task with ample amounts of relevant data. Then, we tune the unfrozen layers to perform well on a related ML task, even if the amount of data acquired for this task is less. As the frozen weights of the architecture are optimized to capture features of a dataset specific to an ML task and have been trained with ample amounts of relevant data for high performance, it transfers its knowledge to improve the performance of other related ML tasks even though the size of the acquired data for these tasks are small.

In section 3.2, we demonstrated the architecture and the results of a DL model trained for optimal performance to classify between L-Alanine and L-Arginine. By showcasing high performance in distinguishing these two amino acid types, this DL model has learnt features specific to identifying amino acids. Therefore, in this section, we investigate if the knowledge that this DL model acquired can be transferred to improve the performance in identifying L-Lysine, which has the least amount of data compared to L-Alanine and L-Arginine. Furthermore, we hypothesized that if this approach works well in identifying L-lysine, we could apply

**Table 7. Accuracy obtained across train, test and validation datasplits in classifying L-Alanine, L-Arginine and L-Lysine through transfer learning approach (Only the final layer was trained).**

| Train | Validation | Test |
|---|---|---|
| 0.61 | 0.54 | 0.60 |

the same technique to identify other amino acid types with relatively fewer absorption profiles than L-Lysine.

To study the capability of utilizing transfer learning to improve the performance of L-Lysine identification, we froze the weights of all the layers in the DL model (used for classifying L-Alanine and L-Arginine) except for the output layer. Then, we set the size of this output layer to 3 (to output a class between L-Alanine, L-Arginine and L-Lysine) and trained it with the acquired data. This enables the output layer to tune its weights with respect to the weights of the frozen layer, which are already tuned to learn the features related to identifying amino acid categories. Tables 7 and 8 demonstrates the performance (in terms of accuracy rounded off to two decimals) of this model in classifying L-Alanine, L-Arginine and L-Lysine. Class weights were incorporated while training, where class weights for each class were calculated as elaborated in the section 3.3. Additionally, since the neural network from which the layers' weights were freezer in our study contains dropout. Therefore, the reported train Acc is the accuracy obtained in the training dataset with the saved best model used to report validation and test Acc. While training, the train Acc obtained was 0.57 (rounded off to two decimals).

It is evident from Table 7 that the model has generalized well to unseen test data. However, its performance in identifying L-Lysine is poor, as demonstrated in Table 8. This behaviour is clearly due to a lack of data. The accuracy scores in the Table 7 clearly demonstrate that the amount of data available for training, testing and validating L-Lysine is insignificant as compared to the other two dominant amino acid categories, which heavily contributed to the higher and generalized accuracy scores. This further reinforces that we could classify other amino acid types if we acquire more data, and the approach to automate the amino acid identification using ML is feasible.

## 4 Conclusions

In this study, we successfully demonstrate the potential of an ML algorithm to precisely identify amino acids through their unique absorption profiles under elliptically polarized light. We proved the efficacy of our approach by successfully classifying between L-alanine and L-arginine with an overall test accuracy of 78%. As the acquired data for the 17 amino acid types were inadequate, the potential of this innovative approach, leveraging meticulous data preprocessing techniques, has been showcased only through the high accuracy in distinguishing amino acid groups, L-Alanine and L-Arginine, which had the highest count of data. Future research will expand the dataset to encompass all amino acid types, addressing current limitations and facilitating comprehensive evaluation. Nevertheless, the findings suggest a significant advancement in biochemical analysis, offering a scalable, efficient alternative to traditional methods. Despite certain limitations, like the skewed distribution of the dataset, this research

**Table 8. Individual amino acid accuracy obtained through transfer learning approach (Only the final layer was trained).**

| L-Arginine | L-Alanine | L-Lysine |
|---|---|---|
| 0.71 | 0.6 | 0.17 |

paves the way for future developments in the field, highlighting the utility of ML in enhancing diagnostic and analytical methodologies in biochemistry and medical diagnostics. In the future, we plan to acquire adequate experimental records for other amino acid types and we plan to deploy the model on the EDS device for real-time automatic amino acid identification.

## Supporting information

**S1 Fig. Stability experiment of ED spectrometer over multiple days.**
(PDF)

**S2 Fig. Baseline corrected and normalized absorption curves.**
(PDF)

## Author Contributions

**Conceptualization:** Tommy Gao, Dharmakeerthi Nawarathna, Umamaheswara Rao Tida, Dali Sun.

**Data curation:** Yusuf Asad.

**Investigation:** Yusuf Asad, Tommy Gao, Umamaheswara Rao Tida, Dali Sun.

**Methodology:** Ridhanya Sree Balamurugan.

**Project administration:** Umamaheswara Rao Tida, Dali Sun.

**Resources:** Ridhanya Sree Balamurugan, Tommy Gao, Dharmakeerthi Nawarathna, Dali Sun.

**Software:** Ridhanya Sree Balamurugan, Dharmakeerthi Nawarathna.

**Supervision:** Umamaheswara Rao Tida.

**Validation:** Ridhanya Sree Balamurugan, Tommy Gao, Umamaheswara Rao Tida, Dali Sun.

**Visualization:** Ridhanya Sree Balamurugan.

**Writing – original draft:** Ridhanya Sree Balamurugan, Tommy Gao, Dali Sun.

**Writing – review & editing:** Ridhanya Sree Balamurugan, Tommy Gao, Dharmakeerthi Nawarathna, Umamaheswara Rao Tida, Dali Sun.

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
