## [Decision Letter · Decision Letter 0]

28 Oct 2024

PONE-D-24-43688Automating the Amino Acid Identification in Elliptical Dichroism Spectrometer With Machine LearningPLOS ONE

Dear Dr. Tida,

Thank you for submitting your manuscript to PLOS ONE. After careful consideration, we feel that it has merit but does not fully meet PLOS ONE’s publication criteria as it currently stands. Therefore, we invite you to submit a revised version of the manuscript that addresses the points raised during the review process.

We look forward to receiving your revised manuscript.

Kind regards,

Saheli Mitra

Academic Editor

PLOS ONE

3. Thank you for stating the following financial disclosure: This work was financially supported by grants from the National Cancer Institute (R21CA270748, R03CA252783) and the National Institute of General Medical Sciences (U54GM128729) of National Institutes of Health to Dali Sun, National Science Foundation CAREER Award (2236885) to Dali Sun and National Science Foundation PFI-TT Award (2300064) to Dharmakeerthi Nawarathna. 

4. Thank you for stating the following in the Acknowledgments Section of your manuscript: This work was financially supported by grants from the National Cancer Institute (R21CA270748, R03CA252783) and the National Institute of General Medical Sciences(U54GM128729) of National Institutes of Health to Dali Sun, National Science Foundation CAREER Award (2236885) to Dali Sun and National Science Foundation PFI-TT Award (2300064) to Dharmakeerthi Nawarathna. 

Please remove any funding-related text from the manuscript and let us know how you would like to update your Funding Statement. Currently, your Funding Statement reads as follows: This work was financially supported by grants from the National Cancer Institute (R21CA270748, R03CA252783) and the National Institute of General Medical Sciences (U54GM128729) of National Institutes of Health to Dali Sun, National Science Foundation CAREER Award (2236885) to Dali Sun and National Science Foundation PFI-TT Award (2300064) to Dharmakeerthi Nawarathna.

Reviewers' comments:

Reviewer's Responses to Questions

**Comments to the Author**

1. Is the manuscript technically sound, and do the data support the conclusions?

Reviewer #1: Yes

Reviewer #2: Yes

2. Has the statistical analysis been performed appropriately and rigorously? 

Reviewer #1: Yes

Reviewer #2: Yes

3. Have the authors made all data underlying the findings in their manuscript fully available?

Reviewer #1: Yes

Reviewer #2: No

4. Is the manuscript presented in an intelligible fashion and written in standard English?

Reviewer #1: Yes

Reviewer #2: Yes

5. Review Comments to the Author

**Reviewer #1: **Based on their previous article, which allows for the characterization of biomolecular structure features with elliptical dichroism spectrometry (EDS), the authors developed a machine-learning method to identify amino acids with the absorption profile. Their results show less prediction power in distinguishing absorption profiles for all 17 amino acids at once but show a better prediction (0.81 for L-Arginine, 0.75 for L-Alanine) when there is a sufficient number of training data (132 absorption profiles for L-Arginine, 142 absorption profiles for L-Alanine).

I like that the authors have an overall good experimental design, and also elaborated the machine learning network structure. I have the following comments:

1) How sensitive/robust is the machine learning identification method? For the proposed methods to work for identifying amino acids in a different solvent, does it need to retrain data for those amino acids in each different solvent? What about varied pH?

2) How noisy is the data after normalization? It would be helpful to include the averaged results from all samples with shadowed IQR or other dispersion ranges.

3) What is the possible source of the noise? Discussing the source of noise between absorption profiles for the same amino acid would be helpful to further improve the prediction accuracy.

4) Would it be possible for the absorption profile to shift left or right along the angle axis due to slightly different sample orientations? If yes, then in data preprocessing, it would be helpful to register the curve according to their patterns

5) On page 5, 3 outliers were thrown out after ranking the Spearman correlation. It makes more sense if outliers are defined by a Spearman correlation threshold. To visualize the distance between all samples, it is helpful to do a PCA analysis and plot in 3D or UMAP.

6) In table 5 and table 7, there is a typo L-L-Lysine.

7) On page 4 table 1, are those absorption profiles obtained from individual samples or repeated measurements?

8) On page 4 line 115, normalizing along each profile is not good for data preprocessing. For example, if two examples have similar line patterns along the angle axis, but different absorption levels, after normalizing by each row, these two patterns will be similar. This will negatively affect the model's predictability. The corresponding code is in code block 6 in Data_Preprocessing/2_ED_paper_amino_AI_normalizationafterbaselineremoval.ipynb

9) In figures, does absorption have units, such as /(M^3cm^2)?

**Reviewer #2: **1. Is the manuscript technically sound, and do the data support the conclusions?

Due to the inadequate data count the results are mainly focused on two type of amino acids. So, it is difficult to generalize the study with inadequate and limited data for other amino acids. Authors may comment on it and mentioned it in the conclusion as future scope of this work.

2. Have the authors made all data underlying the findings in their manuscript fully available?

The absorption profiles for 17 type of amino acids are not added in the manuscript. I have suggested to add a figure with normalized and corrected absorption profiles (with error bars) for all amino acids can be added in the MS to visualize the difference between all of them.

This data can be added in the main MS or as a supporting document.

6. PLOS authors have the option to publish the peer review history of their article (what does this mean?). If published, this will include your full peer review and any attached files.

Reviewer #1: **Yes: **Huijing Wang

Reviewer #2: **Yes: **Prashant Hitaishi

---

## [Author Response · Author response to Decision Letter 0]

26 Nov 2024

Response to Reviewer #1 - Huijing Wang

We thank Huijing for their comments regarding the use of good experimental design and elaboration of the machine learning network. We hope to clarify concerns raised following.

Question 1: How sensitive/robust is the machine learning identification method? For the proposed methods to work in identifying amino acids in a different solvent, does it need to retrain data for those amino acids in each different solvent? What about varied pH?

Response: We appreciate the reviewer’s questions on the sensitivity and robustness of our machine learning method under varying solvent and pH conditions. Blank measurements of the solvent, conducted under identical conditions and replicate numbers as the samples, capture solvent-specific or pH-induced spectral variations. Normalizing sample data with these blanks isolates the intrinsic spectral characteristics of amino acids, facilitating accurate identification without necessitating retraining. This methodology ensures the model's robustness and adaptability across diverse experimental setups. We have elucidated this process in the manuscript.

Question 2: How noisy is the data after normalization? It would be helpful to include the averaged results from all samples with shadowed IQR or other dispersion ranges.

Response: We acknowledge the reviewer’s request for more insights into the noise levels post-normalization. As illustrated in Figure R1, the ED spectrometer maintains a consistent signal-to-noise ratio (SNR) across multiple days, demonstrating robust repeatability. The SNR for our system averages around 14, which, while slightly lower than the SNR of 20 reported for conventional CD spectrometers, remains competitive. This stability underscores the reliability of the ED spectrometer in capturing reproducible spectral profiles.

To elucidate further, the shaded regions in the plot denote the standard error (S.E.), which captures the day-to-day variability while demonstrating the instrument's stability. These findings substantiate that our normalized data exhibits sufficient robustness for precise amino acid identification without significant noise interference. This elucidation and Figure S1 have been incorporated into the manuscript and added to the supporting information to enhance transparency and clarity.

Question 3: What is the possible source of the noise? Discussing the source of noise between absorption profiles for the same amino acid would be helpful to further improve the prediction accuracy.

Response: We appreciate the reviewer’s inquiry regarding the potential sources of noise in our ED spectrometer. The primary source of noise originates from the utilization of a cost-effective photodetector, selected to mitigate the complexity and expense associated with the

photomultiplier tubes (PMTs) employed in conventional CD spectrometers. This design decision, while marginally reducing the signal-to-noise ratio (SNR) to approximately 14 (compared to an SNR of 20 in CD spectrometers), aligns with our objective to simplify the ED spectrometer for enhanced portability and usability. These characteristics are crucial for clinical applications, particularly in cancer detection, where expeditious and accessible diagnostics are imperative. Moreover, with all optical components securely affixed to an aluminum breadboard, external mechanical perturbations are minimized, ensuring stable and reproducible measurements. This equilibrium between simplicity, portability, and adequate sensitivity renders the ED spectrometer an optimal candidate for implementation in clinical environments. These aspects have been incorporated into the revised manuscript for enhanced clarity.

Question 4: Would it be possible for the absorption profile to shift left or right along the angle axis due to slightly different sample orientations? If yes, then in data preprocessing, it would be helpful to register the curve according to their patterns.

Response: We appreciate the reviewer’s insightful question regarding potential shifts in the absorption profile due to sample orientation. This concern is mitigated by several design features of the ED spectrometer. All optical components are securely mounted on an aluminum breadboard, thereby eliminating any mechanical movement within the system. The motorized rotator, which is integral to generating elliptically polarized light, provides precise angle increments, thus ensuring consistent measurements. Furthermore, the cuvette is positioned in a custom 3D-printed holder with tight tolerances, minimizing any variation in sample orientation.

These measures ensure the integrity and reproducibility of absorption profiles, even across multiple measurements.

Question 5: On page 5, 3 outliers were thrown out after ranking the Spearman correlation. It makes more sense if outliers are defined by a Spearman correlation threshold. To visualize the distance between all samples, it is helpful to do a PCA analysis and plot in 3D or UMAP.

Response: We appreciate the reviewer’s insightful suggestion regarding the use of a Spearman correlation threshold for outlier detection and the recommendation to enhance data visualization with PCA or UMAP. While these methodologies can provide additional insights, our current approach, which involves removing the three least-correlated absorption profiles based on their Spearman correlation with the mean sequence, was selected to balance computational simplicity and data quality. This method effectively mitigates the impact of significant outliers without introducing additional complexity, thereby ensuring the robustness of our machine learning model.

We acknowledge the advantages of Principal Component Analysis (PCA) or Uniform Manifold Approximation and Projection (UMAP) in visualizing sample distribution; however, these techniques were not incorporated into our workflow as they are not integral to the primary objective of this study, which focuses on demonstrating the feasibility of amino acid

classification using our ED spectrometer and machine learning. Instead, our preprocessing pipeline, including Spearman correlation-based outlier removal, ensured that the dataset retained its representative diversity while minimizing noise.

We posit that our approach provides a streamlined, effective method for data preprocessing, particularly considering our goal to develop a portable, user-friendly system for clinical applications. However, we appreciate the reviewer's suggestion and will consider incorporating advanced visualization techniques in future studies to further enhance interpretability.

Question 6: In table 5 and table 7, there is a typo L-L-Lysine.

Response: This has been corrected.

Question 7: On page 4 table 1, are those absorption profiles obtained from individual samples or repeated measurements?

Response: We appreciate the reviewer’s inquiry regarding Table 1 on page 4. The absorption profiles presented in the table were derived from individual amino acid samples at different time points.

Question 8: On page 4 line 115, normalizing along each profile is not good for data preprocessing. For example, if two examples have similar line patterns along the angle axis, but different absorption levels, after normalizing by each row, these two patterns will be similar. This will negatively affect the model’s predictability. The corresponding code is in code block 6 in Data_Preprocessing/2_ED_paper_amino_AI_normalizationafterbaselineremoval.ipynb.

Response: We appreciate the reviewer’s insightful observation regarding the normalization process. As elucidated in the manuscript (Section 2.2, lines 115-125), normalization is implemented to amplify variations within each absorption profile and facilitate efficient convergence of neural network weights, which is essential for stable and expeditious training.

Question 9: In figures, does absorption have units, such as /(M^3cm^2)?

Response: We appreciate the reviewer’s question. As elucidated in the manuscript, absorption values are indeed dimensionless and consequently do not possess associated units. This phenomenon is attributed to the fact that absorption is calculated as the logarithmic ratio of incident to transmitted light intensity, resulting in a unitless quantity.

Response to Reviewer #2 – Prashant Hitaishi

We thank Prashant for their comments regarding our research and note that the limited data makes it difficult to generalize the study. We hope to clarify such discrepancy by stating this in our manuscript while also providing additional feedback to comments below.

Comment 1: Due to the inadequate data count the results are mainly focused on two type of amino acids. So, it is difficult to generalize the study with inadequate and limited data for other amino acids. Authors may comment on it and mentioned it in the conclusion as future scope of this work.

Response: We appreciate the reviewer’s observation regarding the limited data and its implications for generalizing the study. The present investigation aims to demonstrate the viability of utilizing absorption profiles from the ED spectrometer, in conjunction with machine learning techniques, for amino acid identification. Owing to the time-intensive nature of data acquisition, the study focused on L-alanine and L-arginine, which provided sufficient data to validate the proposed approach.

This limitation is duly acknowledged in the manuscript, and it has been emphasized in the conclusion that subsequent research will involve expanding the dataset to encompass all amino acid types. This expansion will facilitate a more comprehensive evaluation of the method's generalizability and robustness. The aforementioned clarification has been explicitly incorporated into the manuscript to address this point.

Comment 2: The absorption profiles for 17 type of amino acids are not added in the manuscript. I have suggested to add a figure with normalized and corrected absorption profiles (with error bars) for all amino acids can be added in the MS to visualize the difference between all of them. This data can be added in the main MS or as a supporting document.

Response: This data has been added to the manuscript within the supporting document.

---

## [Decision Letter · Decision Letter 1]

22 Dec 2024

Automating the Amino Acid Identification in Elliptical Dichroism Spectrometer With Machine Learning

PONE-D-24-43688R1

Dear Dr. Umamaheswara Rao Tida,

We’re pleased to inform you that your manuscript has been judged scientifically suitable for publication and will be formally accepted for publication once it meets all outstanding technical requirements.

Kind regards,

Saheli Mitra

Academic Editor

PLOS ONE

Additional Editor Comments (optional):

Reviewers' comments:

Reviewer's Responses to Questions

**Comments to the Author**

1. If the authors have adequately addressed your comments raised in a previous round of review and you feel that this manuscript is now acceptable for publication, you may indicate that here to bypass the “Comments to the Author” section, enter your conflict of interest statement in the “Confidential to Editor” section, and submit your "Accept" recommendation.

Reviewer #2: All comments have been addressed

2. Is the manuscript technically sound, and do the data support the conclusions?

Reviewer #2: Yes

3. Has the statistical analysis been performed appropriately and rigorously? 

Reviewer #2: Yes

4. Have the authors made all data underlying the findings in their manuscript fully available?

Reviewer #2: Yes

5. Is the manuscript presented in an intelligible fashion and written in standard English?

Reviewer #2: Yes

6. Review Comments to the Author

Reviewer #2: Authors have replied to all questions and revised the manuscript accordingly. I recommend this manuscript for publication.

Best wishes

7. PLOS authors have the option to publish the peer review history of their article (what does this mean?). If published, this will include your full peer review and any attached files.

Reviewer #2: **Yes: **Prashant Hitaishi

---

## [Editor Report · Acceptance letter]

8 Jan 2025

PONE-D-24-43688R1 

PLOS ONE

Dear Dr. Tida, 

I'm pleased to inform you that your manuscript has been deemed suitable for publication in PLOS ONE. Congratulations! Your manuscript is now being handed over to our production team.

Kind regards, 

on behalf of

Dr. Saheli Mitra 

Academic Editor

PLOS ONE